# Near-Infrared Light Irradiation of Porphyrin-Modified Gold Nanoparticles Promotes Cancer-Cell-Specific Cytotoxicity

**DOI:** 10.3390/molecules27041238

**Published:** 2022-02-12

**Authors:** Hiromi Kurokawa, Atsushi Taninaka, Toru Yoshitomi, Hidemi Shigekawa, Hirofumi Matsui

**Affiliations:** 1Faculty of Medicine, University of Tsukuba, Tsukuba 305-8575, Japan; hmatsui@md.tsukuba.ac.jp; 2MoBiol Technologies Corporation, Tsukuba 305-0031, Japan; 3Faculty of Pure and Applied Sciences, University of Tsukuba, Tsukuba 305-8573, Japan; jun_t@bk.tsukuba.ac.jp (A.T.); hidemi@bk.tsukuba.ac.jp (H.S.); 4TAKANO Co., Ltd., Nagano 399-4301, Japan; 5Research Center for Functional Materials, National Institute for Materials Science, Tsukuba 305-0044, Japan; yoshitomi.toru@nims.go.jp

**Keywords:** photodynamic therapy, Au-HpD, nearinfrared, reactive oxygen species

## Abstract

The use of nanoparticles has been investigated as a new cancer treatment. These can induce specific cytotoxicity in cancer cells. In particular, Au nanoparticles (AuNPs) have unique characteristics. The maximum absorption spectrum of AuNPs can be adjusted to modify their size or shape to absorb near-infrared light that can penetrate into tissue without photodamage. Thus, the combination of AuNPs and near-infrared light can be used to treat cancer in deep-seated organs. To obtain effective cancer-specific accumulation of AuNPs, we focused on porphyrin and synthesized a porphyrin-attached Au compound: Au-HpD. In this study, we investigated whether Au-HpD possesses cancer-specific accumulation and cytotoxicity. Intracellular Au-HpD accumulation was higher in cancer cells than in normal cells. In order to analyze the cytotoxicity induced by Au-HpD, cancer cells and normal cells were co-cultured in the presence of Au-HpD; then, they were subjected to 870 nm laser irradiation. We observed that, after laser irradiation, cancer cells showed significant morphological changes, such as chromatin condensation and nuclear fragmentation indicative of cell apoptosis. This strong effect was not observed when normal cells were irradiated. Moreover, cancer cells underwent cell apoptosis with combination therapy.

## 1. Introduction

A WHO report shows that the leading cause of mortality worldwide is ischemic heart disease, followed by cancer. The number of new cancer patients in 2018 was calculated to be 18 million. Moreover, it is estimated that cancer will become the leading cause of mortality in 2060. Lung, liver, and stomach cancers are the top lethal cancers in the general population [1]. The traditional and latest treatments for cancer include surgery, chemotherapy, and radiotherapy, but all of these treatments cause major side effects in normal tissues. Moreover, some cancers have better prognoses, while other cancers, such as pancreatic cancer, have worse prognoses. Consequently, the optimal cancer treatment will have high, cancer-specific toxicity and a low side effect profile for normal tissues [2].

The use of nanoparticles, including Au nanoparticles (AuNPs), has been investigated as a new cancer treatment that induces specific cytotoxicity in cancer cells [3]. AuNPs have unique characteristics, such as high biocompatibility, low cytotoxicity for normal tissues, and simple synthesis [4]. Based on these characteristics, many researchers have investigated new cancer therapies using these particles [5,6]. AuNPs enhance reactive oxygen species (ROS) generation in the following two ways: thermal effect and localized surface plasmon resonance [3,4]. The maximum absorption spectrum of AuNPs can be adjusted to modify their size or shape to absorb near-infrared (NIR) light that can penetrate into tissue without photodamage. Thus, the combination of AuNPs and NIR light can be used to treat cancer in deep-seated organs, such as the liver, pancreas, and kidney.

AuNPs are cancer-selective chemo reagents because they induce cancer-specific cytotoxicity. However, enhancement of cancer-specific accumulation is desired to achieve a more effective treatment. To obtain effective cancer-specific accumulation of AuNPs, we focused on photodynamic therapy (PDT), a type of cancer treatment that is influenced by the accumulation of photosensitizers, such as porphyrins [5]. Porphyrin accumulation was shown to be higher in cancer tissues than in normal tissues. When exposed to laser irradiation with the optimal wavelength, porphyrins induce the production of ROS via a photoreaction, causing cancer-specific damage. Cellular uptake of porphyrin is induced by heme carrier protein-1 (HCP-1), whose expression is higher in cancer cells than in normal cells [6,7]. We reasoned that porphyrin attachment could function in the accumulation of AuNPs in cancer cells via HCP-1.

In this study, we synthesized a porphyrin-attached Au compound and found that this compound preferentially accumulates in cancer cells and induces cancer-specific cytotoxicity in combination with laser irradiation to generate the plasmonic effect.

## 2. Results

### 2.1. Preparation and Characterization of Au-HpD

HpD-PEG-SH and FL-PEG-SH were synthesized, and the porphyrin modification in the Au compound, Au-HpD, was prepared by HpD-PEG-SH and FL-PEG-SH (Figure 1). The introduction of HpD into HpD-PEG-SH was confirmed by gel permeation chromatography (GPC) (Appendix A). The elution of polymers with/without fluorescent dye, such as HpD and FL, can be detected by the refractive index detector, while the elution of fluorescent-labeled polymers can be detected by a UV detector. As shown in Appendix A (Appendix A), heterofunctional polyethylene glycol with amine and thiol ends (NH2-PEG-SH) had no absorbance, while HpD-PEG-SH showed strong absorbance at 400 nm at the elution time of HpD-PEG-SH. This result indicates the introduction of HpD into the HpD-PEG-SH. From the GPC chromatogram, the peak of HpD-PEG-S-S-PEG-HpD with a disulfide bond was detected. Therefore, tris (2-carboxyethyl) phosphine solution was used for the reduction of the disulfide bond in the preparation of Au-HpD. In addition, free HpD was detected in Appendix A. Free HpD can be removed in the purification step of Au-HpD. Similarly, the introduction of FL into the FL-PEG-SH was confirmed by GPC (Appendix A). The mean hydrodynamic diameter of Au-HpD was 58.8 nm, and the polydispersity index was 0.185 (Figure 2A). The UV–Vis absorption spectra in PBS are shown in Figure 2B. Au has two bands at 530 nm and 870 nm. Au-HpD also had the same two bands. Thus, the bands of Au-HpD were derived from Au.

### 2.2. Accumulation of Au-HpD Is Higher in Cancer Cells Than in Normal Cells

Taking into consideration that cancer cells incorporate higher levels of porphyrins than normal cells, we analyzed whether Au-HpD has any preferential accumulation in cancer cells. For this, RGK1 and RGM1 cells were incubated in a growth medium with Au-HpD for 24 h. The analysis showed that Au-HpD absorption in RGK1 cancer cells was significantly higher than in RGM1 normal cells (Figure 3). This result suggests that intracellular Au-HpD accumulation in cancer cells was higher than in normal cells.

### 2.3. Au-HpD Induces Cytotoxicity in Cancer Cells

In order to analyze the cytotoxicity induced by Au-HpD, RGK-KO, and RGM-GFP, cells were co-cultured in the presence of Au-HpD for 24 h and then subjected to 870 nm laser irradiation for 30 min. RGM-GFP cells resulted from the transfection of RGM1 cells with a construct expressing a green fluorescent protein (GFP) gene, which fluoresces in green when excited with blue light. RGK-KO cells were established by the transfection of RGK1 cells with a construct expressing the kusabira orange (KO) protein gene [8]. In the entire cell population, RGK cancer cells were distinguishable from RGM normal cells as they express different fluorescent proteins. We observed that, after laser irradiation, RGK cancer cells showed significant morphological changes, such as chromatin condensation and nuclear fragmentation indicative of cell apoptosis (Figure 4A,B,D,E). This strong effect was not observed when RGM normal cells were irradiated (Figure 4A,C,D,F). Cell apoptosis was detected using Hoechst 33342. The phase-contrast image is shown in Figure 5A, and the fluorescence images are shown in Figure 5B–D. Morphological changes indicative of cell apoptosis, such as condensation of both chromatin and nuclear fragmentation, were observed in RGK-KO cancer cells (Figure 5B,C). However, it was not observed in RGM-GFP normal cells (Figure 5B,D). Taken together, our results show that there is increased cytotoxicity associated with the amount of Au-HpD in the cell.

### 2.4. Intracellular ROS Generation Increased in RGK Cells in Combination with Au-HpD and Laser Irradiation

We analyzed the production of intracellular ROS in cells with Au-HpD. We observed that incubation with Au-HpD induced a significant increase in ROS in RGK1 cells, even before they were laser irradiated, when compared to that in RGM1 normal cells. This result indicates that intracellular ROS production in cancer cells treated with Au-HpD is higher than in normal cells treated with the same compound. After irradiation, intracellular ROS increased significantly in RGK1 cancer cells when compared with that in non-irradiated cells. The intracellular ROS levels of RGM1 normal cells treated with Au-HpD were not significantly altered after irradiation (Figure 6).

## 3. Discussion

In this study, we prepared Au-HpD and evaluated its characteristics. Intracellular Au-HpD accumulation in RGK1 cells was significantly higher than in RGM1 normal cells. Moreover, combination treatment with Au-HpD and laser irradiation induced cancer-specific apoptosis and ROS generation.

Nanoparticles have been developed with potential applications as diagnostic and therapeutic agents, and nanoparticle-based therapies are a promising approach to overcome conventional cancer therapies [8]. Magnetic nanoparticles can induce hyperthermia in the presence of an alternating magnetic field [9,10]. In addition, metallic nanoparticles with a high atomic number release Auger electrons in response to X-ray radiation, having the potential to locally enhance the effects of radiation therapy when specifically delivered to tumors [11]. Over the past decades, therapy using AuNPs has been investigated for several diseases, in particular for cancer [12,13]. However, the identification of modifications to achieve cancer-specific delivery is essential to induce more cancer-specific damage with lower effects in normal cells.

In this study, we focused on AuNPs, which have high biocompatibility and absorption in the NIR region. The absorption coefficients of water and hemoglobin are at their lowest levels in the NIR window, enabling light with longer wavelengths to deeply penetrate the tissue [14]. The use of AuNPs with high cancer-specific accumulation allows for cancer treatment in deep tissues. To enhance the cancer-specific accumulation, we selected porphyrins, which are used as photosensitizers in PDT and were shown to accumulate in cancer cells. They can be activated by optimal wavelength laser irradiation, establishing interactions with the surroundings via different pathways [15]. Such interactions generate ROS that, in turn, induce cytotoxicity more prominently in tumor tissue due to cancer-specific porphyrin accumulation.

Au-HpD showed absorbance at 530 nm and 870 nm (Figure 3). We measured the Au-HpD absorption at 870 nm in both rat gastric normal cells and the corresponding cancer cells. Au-HpD accumulation in cancer cells was higher than in normal cells. Au-HpD accumulation was higher than AuNPs (Appendix A). According to the porphyrin attachment, Au-HpD obtained the cancer-specific accumulation.

Considering that ROS are formed when AuNPs are irradiated by an optimal laser wavelength [16,17], we evaluated the cancer-specific cytotoxicity induced by the combination of Au-HpD and laser irradiation. To address this, we co-cultured normal cells and cancer cells and subjected them to the same conditions, recapitulating the tumor treatment conditions. We observed that, after incorporation of Au-HpD, cancer cells that were laser irradiated showed significant morphological alterations indicative of apoptosis. In parallel, normal cells under the same conditions did not show any morphological alterations. These results suggest that the combination treatment (Au-HpD + laser irradiation) induced cancer-specific apoptosis. Although cancer cells and normal cells were very close in the co-culture system, the combination therapy could induce cytotoxicity specifically in cancer cells, which is an encouraging result, with lower side effects in normal tissues. Moreover, this combination therapy used an emission light with an 870 nm wavelength, which can be highly effective in the treatment of cancers in deeper regions of the organism. We also attached the fluorescein in Au-HpD. After Au-HpD exposure and laser irradiation, if cancer cells have no damage, we considered that laser power was too low or that Au-HpD accumulation was not enough. To confirm that Au-HpD accumulated in cancer cells, we attached the fluorescein in Au-HpD. However, after Au-HpD accumulation and laser irradiation, cancer cells showed cytotoxicity. Thus, we did not need confirmation using fluorescein.

Finally, we evaluated intracellular ROS production by the combination therapy. It is already known that intracellular ROS production in cancer cells is higher than that in normal cells [18,19]. AuNPs can induce apoptosis through ROS production [20], and in combination with laser irradiation, they enhance ROS generation via localized surface plasmon resonance [21]. In fact, in our experimental conditions, intracellular ROS production in cancer cells treated with Au-HpD was significantly higher than in normal cells under the same treatment, even before laser irradiation (Figure 6). After laser irradiation, intracellular ROS production in cancer cells increased, while intracellular ROS production in normal cells remained virtually the same as before laser irradiation. AuNPs enhance ROS generation by thermal effects or localized surface plasmon resonance. Au-HpD was excited by NIR light, and thermal or plasmon resonance can enhance ROS production in cancer cells. We discovered that the combination of Au-HpD and laser irradiation induced cancer-specific cytotoxicity and ROS production.

## 4. Materials and Methods

### 4.1. Reagents

AuNPs were purchased from Cytodiagnostics Inc. (Burlington, ON, Canada). HpD was purchased from MedChem Express (San Diego, CA, USA). Fluorescein isothiocyanate (FITC) was purchased from Dojindo (Kumamoto, Japan). A mono-functional polyethylene glycol with thiol end (PEG-SH) and a heterofunctional polyethylene glycol with amine and thiol ends (Mw 2000) were purchased from NOF Corporation (Tokyo, Japan). Dichloromethane (DCM), *N*,*N*-dimethylformamide (DMF), *N*-hydroxysuccinimide (NHS), and *N*,*N′*-diisopropylcarbodiimide were purchased from Fujifilm Wako Pure Chemical (Osaka, Japan) and used with no further purification.

### 4.2. Synthesis of HpD-PEG-SH

Heterofunctional polyethylene glycol with amine and thiol ends (100 mg) was weighed into a 10 mL flask. A mixture of 2 mL of DCM and 2 mL of DMF containing *N*,*N′*-diisopropylcarbodiimide (126 mg), NHS (115 mg), triethylamine (20 mg), and HpD (33 mg) was added to the flask and stirred for 2 h at 25 °C. The HpD-PEG-SH was recovered by precipitation into 40 mL of cold 2-propanol (−15 °C) and centrifugation for 10 min at 5000 rpm (4500× *g*). The resulting PMETAC-co-PAPMAA copolymer was transferred into a pre-swollen membrane tube (Spectra/Por; molecular-weight cutoff size: 1000), dialyzed for 24 h against 2 L of water, which was changed after 2, 5, and 8 h, and then freeze-dried.

### 4.3. Synthesis of FL-PEG-SH

Heterofunctional polyethylene glycol with amine and thiol ends (100 mg) was weighed into a 10 mL flask. A mixture of 2 mL of DMF containing triethylamine (20 mg) and FITC (38 mg) was added to the flask and stirred for 2 h at 25 °C. The FL-PEG-SH was recovered by precipitation into 40 mL of cold 2-propanol (−15 °C) and centrifugation for 10 min at 5000 rpm (4500× *g*). The resulting PMETAC-co-PAPMAA copolymer was transferred into a pre-swollen membrane tube (Spectra/Por; molecular-weight cutoff size: 1000), dialyzed for 24 h against 2 L water, which was changed after 2, 5, and 8 h, and then freeze-dried.

### 4.4. Preparation of Au-HpD

A volume of 10 µL of tris (2-carboxyethyl) phosphine solution (50 mg/mL) was added to 300 µL of PEG stock solution containing 445 µg/mL of PEG-SH, 50 µg/mL of HpD-PEG-SH, and 5 µg/mL of FL-PEG-SH to reduce disulfide bonds. The molar ratio of PEG-SH, HpD-PEG-SH, and FL-PEG-SH was 107: 9:1. This mixture was allowed to react at 25 °C for 30 min and added to 75 µL of AuNPs solution. This mixture was allowed to react at 25 °C for 1 h, followed by centrifugation at 15,000 rpm for 20 min at 20 °C. The obtained solution of AuNPs was centrifuged at 15,000× *g* for 20 min, decanted, and resuspended in PBS to remove excess PEG derivatives. Figure 1 shows a schematic illustration of the Au-HpD. The hydrodynamic diameters of Au-HpD were determined using a Zetasizer Nano (Zetasizer Nanoseries ZEN3600; Malvern Instruments Ltd., Worcestershire, UK). The absorbance of Au-HpD was measured using a Synergy H1 microplate reader (BioTek Instruments Inc., Winooski, VT, USA).

### 4.5. Cell Culture

RGM1 rat gastric mucosal cells were purchased from the RIKEN Cell Bank (Tsukuba, Japan). RGK1 cells were established by exposing RGM1 cells to 1-Methyl-3-nitro-1-nitrosoguanidine. RGM-GFP cells resulted from the transfection of RGM1 cells with a construct expressing a green fluorescent protein (GFP) gene, which fluoresces in green when excited with blue light. RGK-KO cells were established by the transfection of RGK1 cells with a construct expressing the kusabira orange (KO) protein gene [8]. RGM cells were cultured in Dulbecco’s modified Eagles/F12 medium (Life Technologies Japan, Tokyo, Japan). RGK cells were cultured in Dulbecco’s modified Eagles/F12 medium without L-glutamine (Sigma-Aldrich Japan K.K., Tokyo, Japan). These culture media contained 10% heat-inactivated fetal bovine serum (Biowest, Kansas City, MO, USA) and 1% penicillin/streptomycin (Wako Pure Chemical Industries, Osaka, Japan). Cells were cultured in a 37 °C incubator in an atmosphere of 5% CO_2_.

### 4.6. Cellular Uptake of Au-HpD

RGK1 and RGM1 cells were seeded in 12-well plates at a density of 5 × 10^4^ cells/well and incubated at 37 °C for 24 h. Au-HpD was added to cells, and 24 h later, cells were rinsed with phosphate-buffered solution (PBS) and lysed in 100 μL of RIPA buffer. The cell homogenates were transferred to a 96-well plate, and the absorbance was measured at 870 nm using a Synergy H1 microplate reader (BioTek Instruments Inc., Winooski, VT, USA). The protein concentration of each sample was estimated by the bicinchoninic acid protein assay kit (Takara Bio, Shiga, Japan). The absorbance was measured at 562 nm using a Synergy H1 microplate reader.

### 4.7. PDT Using Au-HpD

RGK1 and RGM1 cells were grown overnight in a 60 mm dish and then incubated with Au-HpD for 24 h. After incubation, the medium was changed to fresh medium without phenol red, and cells were irradiated with supercontinuum laser light (870 nm, 1.5 ± 3 mW/cm^2^) using a SuperK FIANIUM FIU-15 (NKT Photonics, Birkerod, Denmark). 

### 4.8. Detection of Apoptotic Cells by Hoechst 33258 Staining

RGK1 and RGM1 cells were cultured in 60 mm dishes and incubated overnight. After PDT treatment, RGK1 and RGM1 cells were incubated with 5 µM Hoechst 33342 for 30 min. Apoptosis, with condensed and fragmented nuclei, was analyzed using the fluorescence microscope IX83.

### 4.9. Intracellular Reactive Oxygen Species (ROS) Measurement

Intracellular ROS were detected using a 2-[6-(4-hydroxy) phenoxy-3H-xanthen-3-on-9-yl] benzoic acid (HPF) (Goryo Chemical, Hokkaido, Japan) fluorescence dye. Nonfluorescent HPF would be *O*-dearylated upon reaction with ROS and then show strong fluorescence. RGK1 and RGM1 cells were cultured in 60 mm dishes overnight. After PDT treatment, RGK1 and RGM1 cells were incubated with 5 μM HPF in MSF buffer (5.4 mM KCl, 136.9 mM NaCl, 8.3 mM glucose, 0.44 mM KH_2_PO_4_, 0.33 Na_2_HPO_4_, 10.1 mM HEPES, 1 mM MgCl_6_ 6H_2_O, 1 mM CaCl_2_ 2H_2_O) for 10 min. After incubation, the solution was replaced with fresh MSF buffer. Fluorescence intensity was analyzed using a fluorescence microscope IX83 (Olympus Optical Co Ltd., Tokyo, Japan). HPF was excited using a 460–495 nm filter, and the emission was collected using a 510–550 nm filter. The protein concentration of each sample was estimated by the bicinchoninic acid protein assay kit (Takara Bio, Shiga, Japan). The absorbance was measured at 562 nm using a Synergy H1 microplate reader.

### 4.10. Statistical Analysis

Data are expressed as the means ± SD and were assessed by an analysis of variance. Individual groups were compared by Tukey’s post hoc or Student’s t-test with *p* < 0.05 considered statistically significant.

## 5. Conclusions

In this study, we describe the use of Au nanoparticles fused with porphyrins in a combination therapy with NIR laser treatment to induce cytotoxicity specifically in cancer and not in normal cells in the same culture. These are encouraging results for the use of this combination therapy in the treatment of different tumors. Additionally, the use of NIR light is expected to induce cancer-specific cell death in deep regions of the organism.

## Figures and Tables

**Figure 1 molecules-27-01238-f001:**
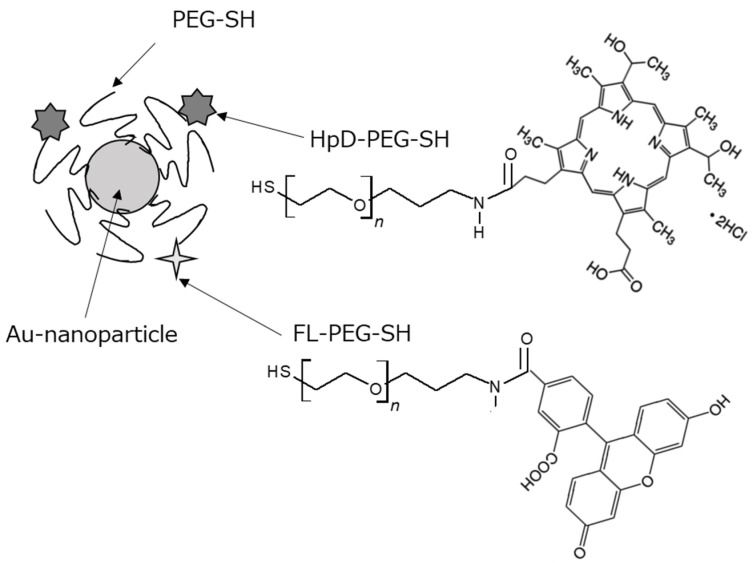
Schematic illustration of the porphyrin-modified Au compound: Au-HpD.

**Figure 2 molecules-27-01238-f002:**
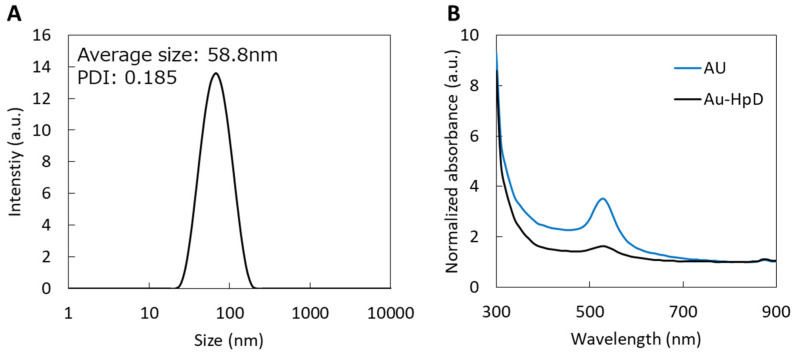
The characteristics of Au-HpD: (**A**) size distributions of Au-HpD; (**B**) UV−Vis absorption spectrum of Au-HpD in a PBS solution.

**Figure 3 molecules-27-01238-f003:**
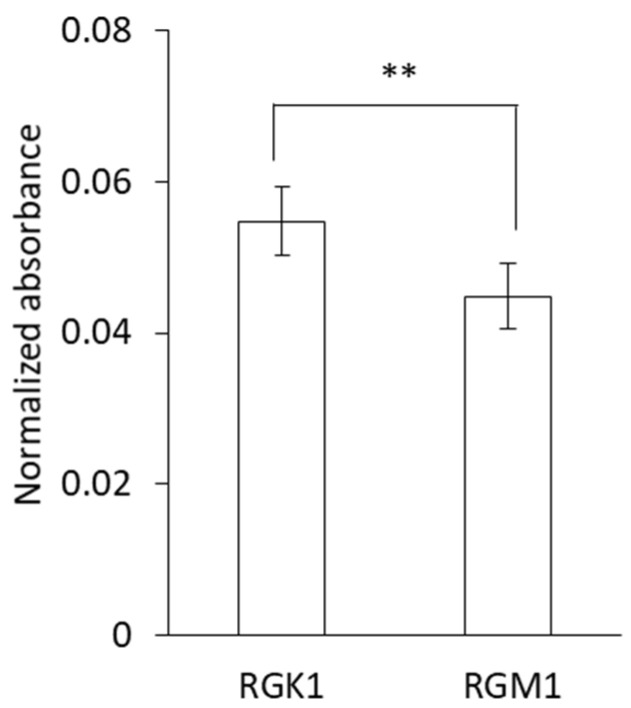
Intracellular Au-HpD accumulation in cancer cells and normal cells. Data are expressed as means ± SD (n = 5). ** *p* < 0.01.

**Figure 4 molecules-27-01238-f004:**
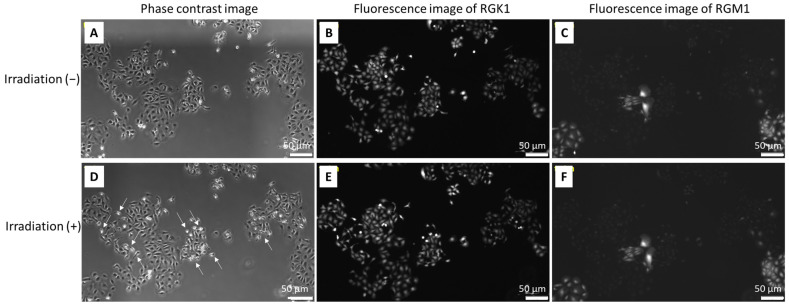
The cytotoxicity combination with Au-HpD and laser irradiation for cells: top row, before laser irradiation; bottom row, after laser irradiation; phase-contrast image (**A**,**D**); fluorescence image of RGK-KO (**B**,**E**); fluorescence image of RGM-GFP (**C**,**F**).

**Figure 5 molecules-27-01238-f005:**
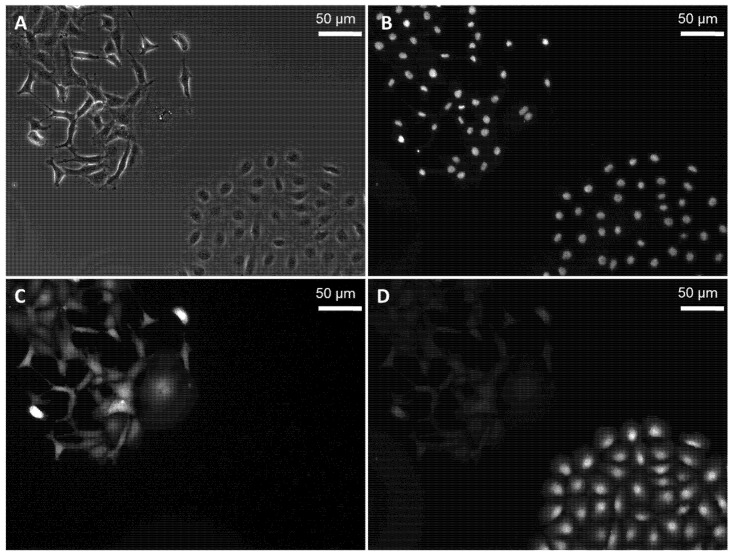
Hoechst 33258 staining of RGK-KO and RGM-GFP cells: phase-contrast image (**A**); fluorescence image of Hoechst 33342 (**B**); RGK-KO (**C**); RGM-GFP (**D**).

**Figure 6 molecules-27-01238-f006:**
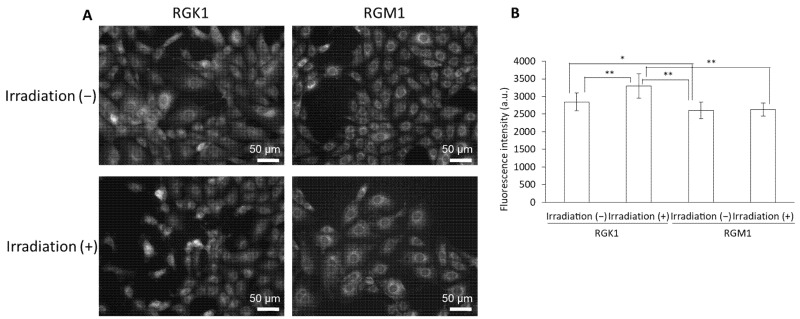
The fluorescence intensity of HPF: (**A**) fluorescent microscopy utilized to assess cellular uptake of HPF; (**B**) data are expressed as means ± SD (n = 20). Ex = 460–495 nm and Em = 510–550 nm. * *p* < 0.05, ** *p* < 0.01.

## Data Availability

Not applicable.

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
