# Peer review of "Near-Infrared Light Irradiation of Porphyrin-Modified Gold Nanoparticles Promotes Cancer-Cell-Specific Cytotoxicity"

_molecules, 2022, doi:10.3390/molecules27041238_

Round 1

Reviewer 1 Report

Kurokawa et al. reported porphyrin-attached AuNPs. The authors claim that these compounds can induce specific cytotoxicity in cancer cells after near-IR light irradiation. This manuscript is well written and the results are nicely laid out. However, this manuscript is not mature yet, and some further results are needed.

  1. Data on the characterization of HpD-PEG-SH and FL-PEG-SH are not available. The authors should provide additional data on characterization such as NMR, mass spectral data, and so on.
  2. Surface morphology and fluorescence imagse (colored) should be compared before and after the AuNPs modified using the chromophores.
  3. The authors should clearly explain why the fluorescein moiety is included and what its role is.
  4. The absorption at 530 nm can be expected to come mainly from the porphyrin moiety, but the absorption at 870 nm in NIR is not obvious. A reasonable explanation based on appropriate evidence is absolutely necessary.
  5. The authors should check the experiments on the cytotoxicity in cancer cells using scavengers. It enlightens more on the nature of the mechanism involving reactive oxygen species (ROS).
  6. Color images are highly recommended in Figures 4, 5, and 6.

Author Response

Reviewer 1

Kurokawa et al. reported porphyrin-attached AuNPs. The authors claim that these compounds can induce specific cytotoxicity in cancer cells after near-IR light irradiation. This manuscript is well written and the results are nicely laid out. However, this manuscript is not mature yet, and some further results are needed.

1. Data on the characterization of HpD-PEG-SH and FL-PEG-SH are not available. The authors should provide additional data on characterization such as NMR, mass spectral data, and so on.

Thank you for your comments. The introduction of HpD and FL into the HpD-PEG-SH and FL-PEG-SH was confirmed by gel permeation chromatography (GPC), respectively. We added the GPC chromatograms in Supporting information (Figures S1 and S2). As shown in Figure S1, hetero-functional polyethylene glycol with amine and thiol ends (NH2-PEG-SH) had no absorbance, while HpD-PEG-SH showed strong absorbance at 400 nm at the elution time of HpD-PEG-SH. This result indicates that the introduction of HpD into the HpD-PEG-SH. From the GPC chromatogram, HpD-PEG-S-S-PEG-HpD with disulfide bond was detected. Therefore, tris(2-carboxyethyl)phosphine solution was used for reduction of disulfide bond in the protocol of preparation of Au-HpD. In addition, free HpD was detected in Figure S2. Yet, free HpD can be removed in the purification step of Au-HpD. Similarly, the introduction of FL into the FL-PEG-SH was confirmed by GPC (Figure S2). We have revised the manuscript to make it easier for the reader to understand.

Figure S1. Gel permeation chromatograms of NH2-PEG-SH and HpD-PEG-SH, which were measured by high-performance chromatography connected to a DP-8020 pump (TOSOH, Japan), a CO-8020 column oven (TOSOH, Japan), a UV-8020 ultraviolet detector at 400 nm (TOSOH, Japan), and an RI-2031 refractive index detector (JASCO, Japan) with Shodex OHpak SB-803HQ columns (Showa Denko, Tokyo, Japan). DMF containing 10 mM LiCl was used as the eluent at a flow rate of 0.5 mL/min at 40°C.

Figure S2. Gel permeation chromatograms of NH2-PEG-SH and FL-PEG-SH, which were measured by high-performance chromatography connected to a DP-8020 pump (TOSOH, Japan), a CO-8020 column oven (TOSOH, Japan), a UV-8020 ultraviolet detector at 480 nm (TOSOH, Japan), and an RI-2031 refractive index detector (JASCO, Japan) with Shodex OHpak SB-803HQ columns (Showa Denko, Tokyo, Japan). DMF containing 10 mM LiCl was used as the eluent at a flow rate of 0.5 mL/min at 40°C.

2. Surface morphology and fluorescence imagse (colored) should be compared before and after the AuNPs modified using the chromophores.

We measured the absorption wavelength before and after modification and there is no difference (Figure 2B). We considered the shape of the spectrum does not change significantly.

3. The authors should clearly explain why the fluorescein moiety is included and what its role is.

After Au-HpD exposure and laser irradiation, if cancer cells have no damage, we consider that laser power is low or Au-HpD accumulation was not enough. To confirm that Au-HpD was accumulated in cancer cells, we attach the fluorescein in Au-HpD. However after Au-HpD accumulation and laser irradiation, cancer cells was induced cytotoxicity. Thus we did not need the confirmation using fluorescein. According to reviewer’s comments, we added this sentences in page 6.

4. The absorption at 530 nm can be expected to come mainly from the porphyrin moiety, but the absorption at 870 nm in NIR is not obvious. A reasonable explanation based on appropriate evidence is absolutely necessary.

In this study, we would like to excite not porphyrin but Au-HpD. Au-HpD has an absorption maximum at 530 nm and 870 nm (Figure 2). We assumed that Au-HpD can treat the cancer in deeper regions and selected 870 nm.

5. The authors should check the experiments on the cytotoxicity in cancer cells using scavengers. It enlightens more on the nature of the mechanism involving reactive oxygen species (ROS).

According to reviewer's comments, combination with antioxidant is very important because we think that the cytotoxicity by Au-HpD and NIR in induced by intracellular ROS generation. The aim of this study is to show that combination therapy with Au-HpD and NIR can induce cancer specific cytotoxicity. The reviewer's comments will be discussed in the future works.

6. Color images are highly recommended in Figures 4, 5, and 6.

We decided that there was no need to add a pseudo color. 

Reviewer 2 Report

In this work, Hiromi Kurokawa, Atsushi Taninaka , Toru Yoshitomi , Hidemi Shigekawa , Hirofumi Matsui described their work and results  about “

Near-infrared light irradiation of porphyrin modified gold nanoparticles promotes cancer cell specific cytotoxicity”.

I will recommend this article for publication, however after some major revisions.

Indeed, the experimental procedures need to be more detailed.

            For example, HPF method for ROS detection is barely introduced in results and discussion (the abbreviation is even used for the first time without any explcitation) which will be not easy to understand for futures readers.

            Concerning the systems studied, the authors did not explain how they can be sure that porphyrin have been grafted efficiently on the PEG? Some physical-chemical characterization after this first step will be a great asset for the publication as the ratio of HpD per Au Nanoparticles.

            Moreover, I don’t understand why the authors have not studied AU NP grafted only with HpD or FL and use them as control. Nothing is telling on the fluorescence of FL in the whole system. I guess the goal here is to visualize the NP in cells but it is not cleary explained and how can the authors be sure that fluorescence of FITC is not different in the NP by comparison of alone? What is the ratio is their system between HpD and FL? Is this ratio reproductible and if not, what is its effect on ROS. Production and accumulation in cancer cells?

            Concerning the absorption profile of the AU-HPD species, it clearly lacks of description: we can see different bands, the authors should attribute them and compare to the profiles obtained for reference compounds.

            Concerning accumulation of AU-HpD, there is no control. what about control AuNP without HpD and HpD without AuNP. Is there a synergetic effect? Are the authors sure (and how) that FL has here no influence? Again, for the ROS detection with HPF, there is no control except the dark one. The same question can thus apply here also, and especially the ratio HpD/Au NP is of upmost importance.

Author Response

 For example, HPF method for ROS detection is barely introduced in results and discussion (the abbreviation is even used for the first time without any explcitation) which will be not easy to understand for futures readers.

> Thank you for your comment. According to reviewer’s comments, we added the sentence in page 8;

Nonfluorescent HPF would be O-dearylated upon reaction with ROS and then HPF show the strong fluorescence.

Concerning the systems studied, the authors did not explain how they can be sure that porphyrin have been grafted efficiently on the PEG? Some physical-chemical characterization after this first step will be a great asset for the publication as the ratio of HpD per Au Nanoparticles.

> The ratio of PEG, HpD and Au is 445:50:75. This ratio describe in materials & methods 4.4.

Moreover, I don’t understand why the authors have not studied AU NP grafted only with HpD or FL and use them as control. Nothing is telling on the fluorescence of FL in the whole system. I guess the goal here is to visualize the NP in cells but it is not cleary explained and how can the authors be sure that fluorescence of FITC is not different in the NP by comparison of alone? What is the ratio is their system between HpD and FL? Is this ratio reproductible and if not, what is its effect on ROS. Production and accumulation in cancer cells?

> The aim of this study is to produce gold particles that can be excited by NIR for cancer-specific accumulation and treatment of deep-seated cancer. In fact, we were able to prepare particles that can induce cancer-specific cytotoxicity. We believe that the evaluation of these particles compared to the control is not the purpose of this study.

After Au-HpD exposure and laser irradiation, if cancer cells have no damage, we consider that laser power is low or Au-HpD accumulation was not enough. To confirm that Au-HpD was accumulated in cancer cells, we attach the fluorescein in Au-HpD. However after Au-HpD accumulation and laser irradiation, cancer cells was induced cytotoxicity. Thus we did not need the confirmation using fluorescein.

The ratio of PEG, HpD and Fl is 445:50:5. This ratio describe in materials & methods 4.4.

Concerning the absorption profile of the AU-HPD species, it clearly lacks of description: we can see different bands, the authors should attribute them and compare to the profiles obtained for reference compounds.

>According to reviewer’s comments, we added the data in Figure 2 and sentence in page 3.

Au has two bands at 530 nm and 870 nm. Au-HpD also has same two bands. Thus the bands of Au-HpD were derived Au.

>We revise the sentence in materials & method 4.4

Concerning accumulation of AU-HpD, there is no control. what about control AuNP without HpD and HpD without AuNP. Is there a synergetic effect? Are the authors sure (and how) that FL has here no influence? Again, for the ROS detection with HPF, there is no control except the dark one. The same question can thus apply here also, and especially the ratio HpD/Au NP is of upmost importance.

> In this study, we designed an Au-HpD that can be excited by NIR and can achieve cancer specific accumulation to target deep regio-seated cancer organs. Indeed, combination with Au-HpD and NIR can induce the cancer specific cytotoxicity. Regarding the HPF data, we observed an increase in the production of ROS in cancer cells that had taken up a large amount of Au-HpD. The ratio HpD/Au NPs that increase ROS production will be discussed in the future works.

Round 2

Reviewer 1 Report

Now I recommend the publication of this manuscript as stands.

Author Response

Thank you for your review.

Reviewer 2 Report

I thank the authors to take account of my remarks.

Author Response

Thank you for your review.